# Non-equilibrium properties of an active nanoparticle in a harmonic potential

Falko Schmidt [1], Hana Šípová-Jungová [2], Mikael Käll [2], Alois Würger [3✉] & Giovanni Volpe [1✉]

Active particles break out of thermodynamic equilibrium thanks to their directed motion, which leads to complex and interesting behaviors in the presence of confining potentials. When dealing with active nanoparticles, however, the overwhelming presence of rotational diffusion hinders directed motion, leading to an increase of their effective temperature, but otherwise masking the effects of self-propulsion. Here, we demonstrate an experimental system where an active nanoparticle immersed in a critical solution and held in an optical harmonic potential features far-from-equilibrium behavior beyond an increase of its effective temperature. When increasing the laser power, we observe a cross-over from a Boltzmann distribution to a non-equilibrium state, where the particle performs fast orbital rotations about the beam axis. These findings are rationalized by solving the Fokker-Planck equation for the particle's position and orientation in terms of a moment expansion. The proposed self-propulsion mechanism results from the particle's non-sphericity and the lower critical point of the solution.

[1] Department of Physics, University of Gothenburg, SE-41296 Gothenburg, Sweden. [2] Department of Physics, Chalmers University of Technology, SE-41296 Gothenburg, Sweden. [3] Laboratoire Ondes et Matière d'Aquitaine, Université de Bordeaux & CNRS, F-33405 Talence, France. ✉email: alois.wurger@u-bordeaux.fr; giovanni.volpe@physics.gu.se

Active matter is constituted by particles that can self-propel and, therefore, feature properties and behaviors characteristic of systems that are out of thermodynamic equilibrium[1]. Active-matter systems range across scales going from large robots and animals, down to single-celled organisms and artificial active particles[2–5]. They have found a broad range of applications, e.g., enhancing self-assembly, bioremediation, and drug-delivery[6,7].

The presence of confinement, boundaries, and obstacles has an important influence on the behavior of active particles. For example, motile bacteria form spiral patterns when confined in circular wells[8] and Janus particles reorient at walls[9]. Confinement can be provided also by the presence of external potentials, e.g., electric, magnetic, or chemical potentials. A paradigmatic example of a confining potential is provided by the harmonic potential, which is widely employed to study physics, in general, and thermodynamics, in particular. It can also provide important insight into active-matter systems. Experimentally, the motion of active particles in harmonic potentials has already been studied using macroscopic toy robots walking in a parabolic potential landscape[10], as well as microscopic active colloidal particles in an acoustic trap[11], in an active bath[12–14], and in an optical trap[15]. All these experiments have been performed with relatively large particles, where, in particular, active motion is mainly determined by the particle's self-propulsion, while the particle's rotational diffusion occurs on much longer time scales.

Moving down to the nanoscale, rotational diffusion acquires a much more important role, hindering directed motion[16]. This is because of the different scaling of translational and rotational diffusion: considering a spherical particle of radius $a$, its translational diffusion scales with its linear dimension (i.e., proportional to $a^{-1}$), while its rotational diffusion scales with its volume (i.e., proportional to $a^{-3}$). This limits the possibility of achieving and studying directed active motion on the nanoscale. In fact, while several nanomotors have been proposed and experimentally realized[4,17–20], their activity translates into a hot Brownian motion, i.e., into a higher effective temperature when exploring a potential well[21].

Here, we demonstrate an experimental system where an active nanoparticle held in a potential well features far-from-equilibrium behavior beyond hot Brownian motion. Specifically, we consider a nanoparticle immersed inside a critical binary mixture and confined by the optical potential created by an optical tweezers. At low laser power, the nanoparticle explores the optical tweezers potential as a hot Brownian particle, which is characterized by a Gaussian position distribution given by the Boltzmann factor of the potential. Increasing the laser power, we observe a transition towards a state with a clear out-of-equilibrium signature, where the nanoparticle moves away from the trap center acquiring a non-Gaussian position distribution. Furthermore, the nanoparticle performs orbital rotations around the trapping beam, whose direction we can statistically control by adjusting the polarization of the beam. We provide a theoretical model based on the solution of a Fokker-Planck equation in terms of a moment expansion, which provides strong evidence that the behavior of the nanoparticle in the optical trap is a result of its non-spherical shape. These results demonstrate the importance of asymmetry in nanoscale active systems as a determinant of their behavior in confinement. This insight provides a crucial stepping stone towards the next generation of fast and efficient nanomotors.

## Results

**Experimental setup.** We investigate the dynamics and probability distribution of gold nanoparticles trapped in a focused laser beam ($\lambda = 785$ nm). We employ commercially available monodisperse nanoparticles with radius $a = 75$ nm (Sigma Aldrich, < 12% variability in size). Although often referred to as nanospheres, these nanoparticles feature a crystalline structure that distinguishes them from an ideal sphere, as can be seen in the SEM image in Fig. 1a.

As schematically shown in Fig. 1b, the trapping beam propagates upwards and is focused near the top cover glass surface of the sample cell. The nanoparticle is confined along the vertical $z$-direction at distance $d$ from the cover glass by counteracting actions of the radiation pressure pushing it towards the cover glass and of the short-range electrostatic repulsion pushing it away from the glass surface[22]. Therefore, the nanoparticle is effectively confined in a quasi-two-dimensional space in the $xy$-plane parallel to the cover glass, where it is trapped by an optical tweezers in a harmonic optical potential, i.e., $V(r) = -V_0 e^{-\frac{1}{2}r^2/\sigma^2}$, where $r = \sqrt{x^2 + y^2}$, $\sigma$ is the beam waist and where the prefactor $V_0 = KP$ is proportional to the power $P$ by the proportionality constant $K$.

A schematic of the experimental setup is shown in Supplementary Fig. 1. The nanoparticle motion is captured via digital video microscopy at 719 frames per second.

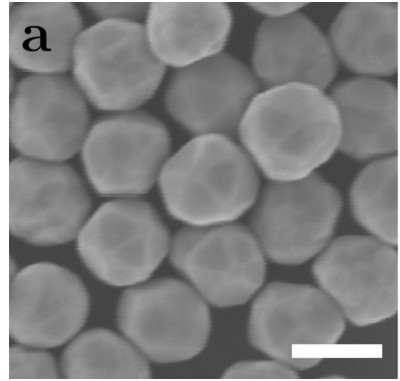
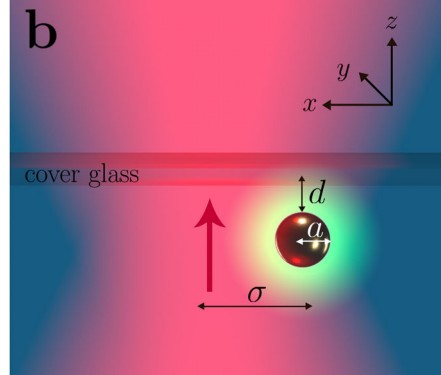

**Fig. 1 Nanoparticles and their driving mechanism. a** SEM image of the gold nanoparticles employed in this work. From this image, it can be appreciated how they are approximately spherical, but also feature characteristic crystalline facets. The scale bar is 150 nm. **b** A nanoparticle (radius $a$) is trapped in a harmonic potential by a focused laser beam (magenta shading, propagating upwards in the direction of the red arrow, beam width $\sigma$). The particle is confined in a quasi-two-dimensional space in the $xy$-plane near the sample upper cover glass at distance $d$ by the competing effects of the optical scattering force and the electrostatic repulsion by the glass. Depending on its distance from the trap center, the nanoparticle is irradiated by different intensities and, therefore, reaches different temperatures $T$. If $T$ exceeds the critical temperature $T_c$, a concentration gradient is locally induced around the nanoparticle (green ring surrounding the particle), thereby leading to a drift velocity away from the trap's center.

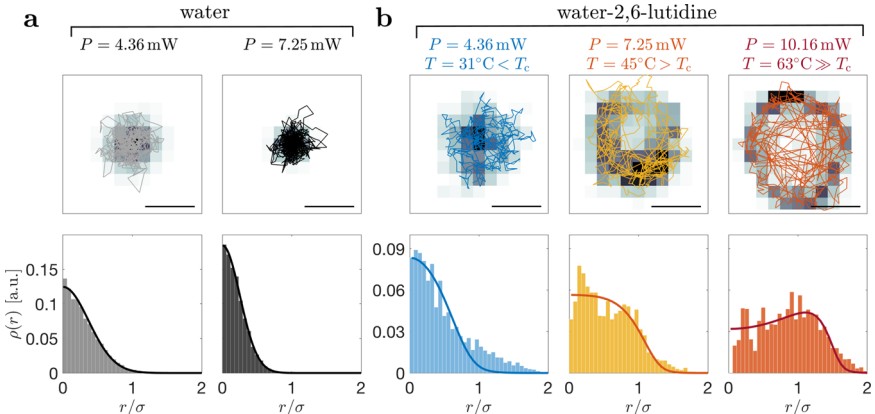

**Fig. 2 Nanoparticle trajectories and probability density distributions.** Trajectories (top) and probability density $\rho(r)$ (bottom) for a nanoparticle with radius $a = 75$ nm held by an optical tweezers, **a** in water at powers $P = 4.36$ (gray), and 7.25 mW (black), and, **b** in a critical mixture of water-2,6-lutidine at $P = 4.36$ (blue), 7.25 (yellow), and 10.16 mW (orange). Sample trajectories in the $xy$-plane are shown for 200 ms, while the background shading indicates the counts (darker colors indicate higher counts); the scalebar is 1 $\mu$m. The probability densities $\rho(r)$ are calculated from data acquired for 1 s. **a** In water, the data are well described by the Boltzmann distribution $\rho_{eq}(r) \propto \exp\left(-\frac{V}{k_B T}\right)$ (solid lines), which becomes narrower as the laser power $P$ and, therefore, the optical trap potential depth increase. **b** In water-2,6-lutidine, the particle features an out-of-equilibrium distribution, which broadens with increasing laser power. Here, the solid lines are given by Eq. (5). All data are taken at environment temperature $T_0 = 3$ K, i.e., $\approx 30$ K below $T_c$. Due to absorption, the particle's surface temperature increases by 6 K mW$^{-1}$ such that, at $P = 7.25$ and 10.16 mW, $T$ exceeds $T_c$ and the solution locally demixes. The radial distance has been normalized by the beam waist $\sigma = 340$ nm.

**Non-equilibrium state**. We start by trapping the particle in water to establish a baseline in a standard medium[23]. The trajectories and the resulting probability density histograms at laser power $P = 4.4$ and 7.3 mW are shown in Fig. 2a. The data are fitted with the Boltzmann probability density $\rho_{eq} \propto \exp\left(-\frac{V}{k_B T}\right)$. The particle is confined at the center of the beam, where the potential may be replaced by its harmonic approximation $V_h = V_0 r^2/\sigma^2$. Indeed, the data in Fig. 2a are very well described by a Gaussian profile. Since the stiffness of the potential increases with laser power, the distribution function is narrower at larger $P$; consequently, an even narrower distribution function is expected at larger laser powers (e.g., $P = 10.16$ mW).

We then study a nanoparticle in a near-critical mixture of water and 2,6-lutidine at a critical lutidine mass fraction $c_c = 0.286$ with a lower critical point at the temperature $T_c \approx 34$ °C (see phase diagram in Supplementary Fig. 2)[24]. At a temperature $T$ below $T_c$ the mixture is homogeneous and behaves as a standard viscous fluid (just like water). When $T$ approaches $T_c$ density fluctuations emerge, leading to water-rich and lutidin-rich regions. Finally, when $T$ exceeds $T_c$ the solution demixes into water-rich and lutidin-rich phases.

Absorption of part of the laser light of the trapping beam heats the nanoparticle and results in a temperature profile in its vicinity. If the surface temperature exceeds $T_c$, a critical droplet with a modified water content $\Phi(r)$ forms around the nanoparticle. Its excess surface temperature is proportional to the laser power. By choosing the critical temperature $T_c$, attained at the critical power $P_c$, as a reference point, the excess temperature can be written as

$$T(r) - T_c = \frac{a^2 \beta}{3\kappa}(Pg(r) - P_c), \quad (1)$$

with the beam profile $g(r) = e^{-\frac{r^2}{2\sigma^2}}$, the absorption coefficient $\beta$, the heat conductivity of the liquid $\kappa$, the laser power $P$, and the critical value $P_c$ corresponding to the laser power at which $T_c$ is attained. For a nanoparticle of $a = 75$ nm, the increase in surface temperature is about 6 K mW$^{-1}$, when the particle is in the highest-intensity region.

In Fig. 2b, we show the probability densities for a nanoparticle trapped at three different laser powers in a near-critical mixture kept at $T_0 = 3$ °C via a heat exchanger coupled to a water bath (i.e., about 30 K below $T_c$). At low laser power ($P = 4.36$ mW, $T = 31$ °C $< T_c$), the nanoparticle position distribution is qualitatively similar to that of the nanoparticle in water (Fig. 2a) and features only very small deviations from a Gaussian profile. As we raise the laser power ($P = 7.25$ mW, $T = 45$ °C $> T_c$), the nanoparticle position distribution acquires a distinctively non-Gaussian shape. Finally, as we raise the laser power even further ($P = 10.16$ mW, $T = 63$ °C $\gg T_c$), the nanoparticle position distribution develops a peak at a finite radial distance $r$ from the trap center, which is also observed in the form of a ring in the histogram of the trajectories. These non-Gaussian distributions cannot be ascribed to a harmonic potential at higher effective distribution and are clear signatures of the out-of-equilibrium nature of this system.

**Self-propulsion of near-spherical particles**. Figure 3 shows the velocity profile $v(r)$ as a function of the distance from the beam axis, as well as its radial and azimuthal components $v_r$ and $v_\theta$. We have determined the local average velocity of the particle by dividing the distance between two subsequent positions by the time separation $\Delta t = 1.39$ ms. This local average velocity consists of an active contribution $u(r)$ depending on the beam intensity and thus on position, and a diffusive contribution $v_D$ that accounts for Brownian motion as well as other random motion components,

$$v(r) = \sqrt{u(r)^2 + v_D^2}. \quad (2)$$

With increasing power, the particle's surface temperature exceeds the lower critical point $T_c$ of water-2,6-lutidine (see Supplementary Information), causing a local modification of the composition according to the spinodal line of the phase diagram. Then, the particle is surrounded by a droplet of modified water content $\phi(\mathbf{r}) - \phi_c \propto \pm \sqrt{T(\mathbf{r}) - T_c}$, where the sign of the excess term depends on the wetting properties of the surface. Within this droplet, isotherms correspond to iso-composition surfaces.

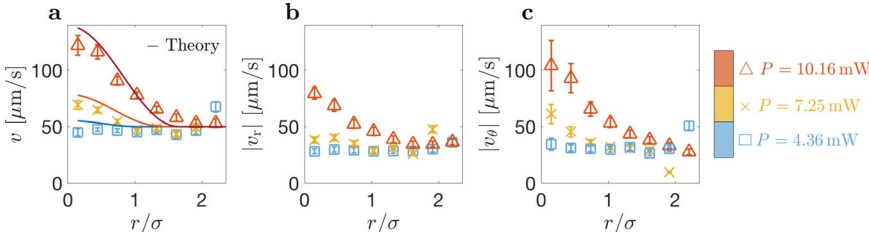

**Fig. 3 Particle velocity dependence on radial position and laser power. a** The particle's total velocity $v$ follows the intensity profile of the laser beam indicated by solid lines given by Eq. (2), where $u_0 = 23.7$, $60.5$ and $131.6\ \mu m\ s^{-1}$ for $P = 4.36$ (blue, square markers), 7.25 (yellow, x markers), and 10.16 mW (orange, triangle markers), respectively, taken from fits of $\rho(r)$ in Fig. 2. Similarly, **b** the absolute radial velocity $|v_r|$ and, **c** the absolute azimuthal velocity $|v_\theta|$ follow the intensity profile of the beam. Data is taken from a single 1-s trajectory sampled at 719 Hz. Each data point is an average over the times the particle passes through that value of $r/\sigma$. Error bars are the standard error of the mean.

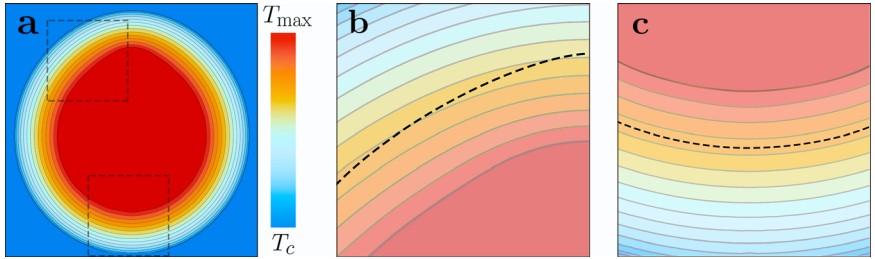

**Fig. 4 Isothermals around a non-spherical particle. a** Composition profile $\phi(\mathbf{r})$ in the vicinity of an axisymmetric particle with a surface temperature above the critical value $T_c$ of water-2,6-lutidine. $\phi$ is constant at the isothermal surface and decreases with distance; the dark blue area indicates the range where $T \leq T_c$ and where the composition takes the bulk value $\phi_c$. The gray lines in the critical droplet ($T \geq T_c$) indicate iso-compositon surfaces. **b** The curvature of the top of the particle is larger than that of its bottom; as a consequence, $\phi$ varies more rapidly close to the top and the iso-composition lines are denser. The dashed line, at a constant distance from the particle surface, crosses iso-composition lines; thus there is a composition gradient $\nabla_\parallel \phi$ parallel to the surface, which induces a diffusio-osmotic creep velocity and results in self-propulsion of the particle. Our detailed analysis relates the particle velocity to the Fourier series of the particle shape $R(\theta)$. **c** Instead, the bottom of the particle is almost spherical with roughly constant curvature and zero creep velocity.

For non-uniformly heated particles, the resulting composition gradient parallel to the surface, $\nabla_\parallel \phi$, drives self-diffusiophoresis. Indeed, active motion above $T_c$ has been reported for both laser-heated Janus particles[25,26] and silica colloids with iron-oxide inclusions[15]. This mechanism was worked out in detail by analytical theory[27] and simulations[28].

Yet, the usual mechanism of self-diffusiophoresis does not apply to homogeneous colloidal spheres, since their symmetry does not allow for a composition gradient along the surface. Therefore, we propose self-propulsion that arises from the non-spherical shape of our nanoparticles, visible in Fig. 1a. The large thermal conductivity of gold imposes an isothermal surface, yet the temperature gradient varies with the local curvature. Thus above the critical point, the composition $\phi(\mathbf{r})$ varies at a constant distance along the particle surface, and the parallel component of the gradient $\nabla_\parallel \phi$ induces creep flow and self-propulsion of the particle. This is schematically shown in Fig. 4, which shows the isothermals (gray lines) surrounding an asymmetric nanoparticle. Moving at a finite distance away from the surface close to an edge (black dashed line, Fig. 4c), multiple isothermals are crossed, indicating a tangential concentration gradient responsible for the nanoparticle motion. For a spherical particle (black dashed line, Fig. 4c) isolines follow the shape of the particle and no tangential concentration gradient is produced. Similar observations have been made for a Leidenfrost ratchet[29].

Starting from an axisymmetric profile $R(\theta) = a(1 + \chi(\theta))$ with $\chi = \sum_n \alpha_n P_n(\cos\theta)$, with the polar angle $\theta$ and Legendre polynomials $P_n$, and evaluating the temperature profile in the vicinity of the isothermal surface of a gold particle, we obtain self-diffusiophoresis at a velocity $u \propto \alpha^2 = \sum_{n=2}^{\infty} \frac{3n+2}{2n+3} \alpha_n \alpha_{n+1}$. Thus,

motion arises from the superposition of odd and even Fourier components of the particle shape. The series starts at $n = 2$, since the dipolar term $n = 1$ corresponds to an irrelevant displacement. For our fits, we assume that less than eight modes contribute with $\alpha_n \sim 0.1$ and thus find $\alpha^2 \sim$ a few percent. For later convenience, we rewrite the self-propulsion velocity as

$$u(r) = \begin{cases} u_0 \dfrac{Pg(r) - P_c}{P - P_c} & \text{for } r < r_c, \\ 0 & \text{for } r > r_c, \end{cases} \qquad (3)$$

with $u_0 = C(P - P_c)$. Note that the velocity depends on the particle position with respect to the beam axis. At a critical distance $r_c = \sigma\sqrt{2\ln P/P_c}$ ($r_c = 570$ nm with $P = 10.16$ mW and $P_c = 2.5$ mW), the local beam intensity is identical to the critical value $P_c$, and the velocity vanishes. For $r > r_c$, the particle is passive. With $C = 12.7\ \mu m\ s^{-1} mW^{-1}$ (in qualitative accord with system parameters, see SI), this expression agrees rather well with the observed dependencies on position $r$ and laser power $P$ (solid lines in Fig. 3a).

As alternative mechanisms, we have also evaluated (and excluded) diffusiophoresis due to the intensity gradient of the laser beam, and spontaneous symmetry breaking due to a small molecular Péclet number. Spontaneous symmetry breaking is excluded since it works only if activity and mobility, as defined in ref. [30], carry opposite signs. This condition can be met by chemically active particles producing a solute that is repelled from the surface, but not by phase separation above a lower critical point because the particle motion tends to diminish the composition gradient along its surface, independently of the wetting properties, while the spontaneous symmetry breaking

would require that the moving particle enhances the gradient in the interaction layer. As to motion driven by the intensity gradient, it is not compatible with the fast orbital motion shown by the trajectories in Fig. 2, nor with the fast motion at the beam center where the gradient vanishes. Details are given in the SI.

Finally, we briefly discuss the anisotropy of the velocity data shown in Figs. 3b and c ($|v_\theta| > v_r$), which is also visible in the trajectories in Fig. 2b. Qualitatively, this is accounted for by the quadrupolar order parameter **Q** (see methods, Eq. (23)). Retaining only the dominant term results in the estimate

$$v_r^2 - v_\theta^2 \sim \frac{u^4}{\sigma^2 D_r^2} \frac{V}{k_B T}. \qquad (4)$$

Because $V < 0$, we find that the mean square of the tangential velocity component exceeds that of the radial one, in agreement with the experiment. Such a velocity anisotropy has been observed previously for a walking robot in a parabolic dish[10]. This effect is readily understood by noting that the radial velocity scale is given by the slow uphill motion, whereas in tangential direction the particle moves at its full speed.

**Probability density and polarization.** The observed probability densities in water–2,6-lutidine shown in Fig. 2b cannot be described by the Boltzmann distribution. In order to relate these deviations to the particle's activity, we have investigated the dynamical behavior in terms of the steady-state distribution $\Psi(\mathbf{r}, \mathbf{n})$, accounting for the gradient diffusion $-D \nabla \Psi$ with Einstein coefficient $D$, the optical tweezers force $\mathbf{F} = -\nabla V$, and the self-propulsion velocity $\mathbf{u} = u\mathbf{n}$. Since the direction of the latter is given by the nanoparticle axis $\mathbf{n}$, the distribution function $\Psi(\mathbf{r}, \mathbf{n})$ depends both on the nanoparticle position $\mathbf{r}$ and on its orientation $\mathbf{n}$, and the Fokker–Planck equation (see methods, Eq. (13)) accounts for rotational diffusion, with coefficient $D_r$, and eventually for spinning motion due to an external torque.

Following previous work on the dynamics of Janus particles[31,32], we resort to a moment expansion $\Psi = \rho + \mathbf{n} \cdot \mathbf{p} + \ldots$, where the probability density $\rho(\mathbf{r}) = \langle \Psi \rangle_\mathbf{n}$ and the polarization density $\mathbf{p}(\mathbf{r}) = \langle \mathbf{n} \Psi \rangle_\mathbf{n}$ are orientational averages with respect to $\mathbf{n}$. When truncating higher-order terms, one readily integrates the steady state

$$\rho(r) \propto \frac{1}{\sqrt{\mathcal{D}^2 + u(r)^2}} \exp\left(-\frac{V}{k_B T} \Phi(r)\right), \qquad (5)$$

where we have defined $\mathcal{D} = \sqrt{6 D_r D}$ and

$$\Phi(r) = \frac{\mathcal{D}}{u_c + u} \arctan \frac{\mathcal{D}^2 - u_c u}{\mathcal{D}(u_c + u)}, \qquad (6)$$

with the shorthand notation $u_c = C P_c$. At the critical radius $r_c$, the velocity $u$ vanishes, and the probability density $\rho(r)$ smoothly reduces to the Boltzmann distribution $\rho_{eq} \propto e^{-V/k_B T}$. With the relation for the bulk diffusion coefficients, $D_r = \frac{3}{4} D/a^2$, the ratio $u/\mathcal{D}$ reduces to the Péclet number $\mathrm{Pe} = \sqrt{2/3} ua/D$, which still depends on position and vanishes at $r = r_c$. The solid curves in Fig. 2b are calculated using Eq. (5), where the optical tweezers potential $V_0 = KP$ is parameterized by $K = 2.97 \times 10^{-17} \mathrm{J\,W^{-1}}$ (corresponding to about $7 k_B T_c$ per 1 mW), whereas the solid curves in Fig. 3a are calculated using Eq. (2) where the velocity is parameterized by $C$ and $P_c$. The fit curves describe the non-equilibrium behavior rather well, and account for the broadening of the distribution and for the bump emerging at $r \approx \sigma$.

Such fits have been done for three different particles at five values of the laser power $P$. Their propulsion speed $u_0$, plotted in Fig. 5, agrees well with Eq. (3). The three particles have the same radius $a$ and absorption coefficient $\beta$; accordingly, they

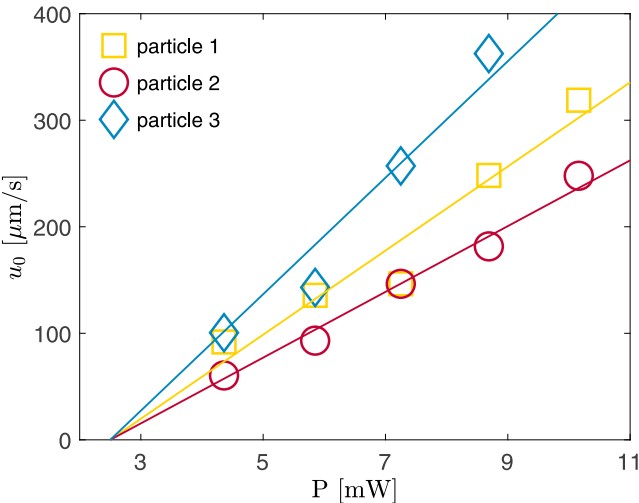

**Fig. 5 Propulsion velocity as a function of the laser power.** The values of the propulsion velocity $u_0$ as a function of the laser power $P$ are obtained from fits like those shown in Fig. 2b, using Eq. (5). The solid line is given by $u_0 = C(P - P_c)$, with $P_c = 2.5$ mW and $C = 39.5$, 30.9, and 54.7 ms$^{-1}$ mW$^{-1}$ for particles 1 (yellow), 2 (red), and 3 (blue), respectively. The data of Figs. 2 and 3 are for particle 1.

experience the same optical tweezers potential and reach the critical point at the same laser power $P_c$ (obtained from the fit of $u_0$ using Eq. (5)). Not surprisingly, the values of the slope $C$ differ significantly, which can be related to the fact that $C$ is proportional to the nonsphericity parameter $\alpha^2$, which varies from one particle to another (see Fig. 1a).

The quantity $\mathcal{D}$ has been calculated with a diffusion coefficient $D$ fitted at low power $P = 4.36$ mW and on time scales larger than the inertial regime (where $\tau \ll 1 \mu s$) of the trajectory mean-squared displacement between 1–20 ms which we found to be linear (see Supplementary Fig. 4). Its value ($D = 1.09 \mu m^2 s^{-1}$) is smaller than the bulk value in water–2,6-lutidine ($D_0 = 2.3 \mu m^2 s^{-1}$, with viscosities taken from ref. [24]). Similarly, the rotational diffusion coefficient used for the fitted curves of Figs. 2 and 5 is smaller than the theoretical value. There are two physical mechanisms which are probably at the origin of this discrepancy: hydrodynamic coupling close to a solid boundary and the confining effect of the critical droplet surrounding an active particle heated above $T_c$. The former reduces the drag coefficient of a sphere moving parallel to a wall[33]. For the latter, the critical droplet formed locally around the particle does not follow its motion but lags behind thus slowing down the particle's diffusion. A more detailed discussion is found below.

**Controlling the direction of orbital rotation.** Transfer of angular momentum from circularly polarized laser light to plasmonic nanoparticles is an efficient means for fueling nanoscopic rotary motors at high-spin rates[34]. It has already been shown theoretically and experimentally verified that, even in a tightly focused Gaussian beam with circular polarization, spin-to-orbital light momentum conversion occurs and can lead to effects such as orbit splitting[35–37]. Here, we show that the spinning motion of an active particle results in orbital trajectories whose preferred handedness is imposed by the polarization of the beam. These measurements are taken with gold nanoparticles of $a = 100$ nm, at $P = 1$ mW, and at room temperature, thus leading to an increase in surface temperature of about 40 K, corresponding to 30 K above $T_c$.

We have investigated the azimuthal component of the velocity depending on the polarization of the beam (Fig. 6a–c). For

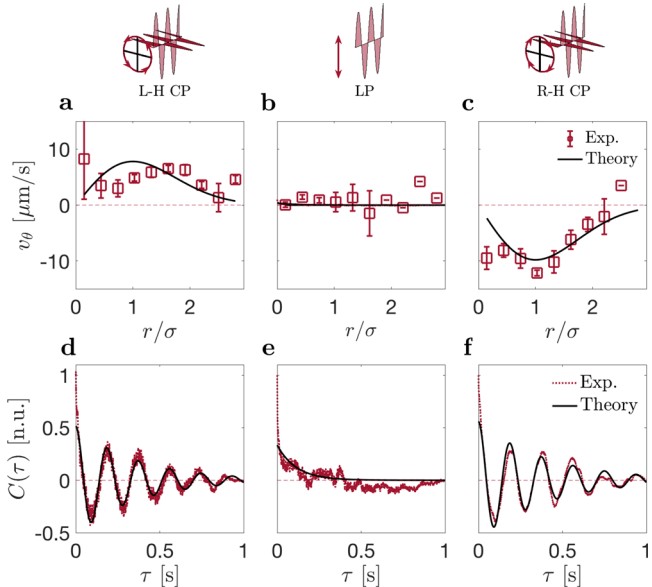

**Fig. 6 Controlling the direction of orbital rotation through light polarization.** The particle orbital rotation is biased toward the direction of the polarization of the trapping beam (laser power $P = 1$ mW, nanoparticle radius $a = 100$ nm). **a–c** Experimental values (red symbols) and theoretical fits (black lines) of the azimuthal velocity $v_\theta$: **a** for left-handed circular polarization, $v_\theta > 0$ showing counter-clockwise orbital rotation of the particle; **b** for linear polarization (see also Figs. 1 and 2), $v_\theta \approx 0$ showing no preferred direction of rotation; and, **c** for right-handed circular polarization, $v_\theta < 0$ showing clockwise orbital rotation. Error bars are the standard error of the mean. The solid line is calculated from Eq. (7) with $\Omega$ the same as in **a–c**, and taking $D_r = 70$ s$^{-1}$, $K = 1.27 \times 10^{-16}$ J W$^{-1}$ (corresponding to about 30 $k_B T_c$ per 1 mW), $u_0 = 120$ $\mu$m s$^{-1}$, and $P_c = 0.2$ mW. **d–f** Experimental (red lines) and theoretical fits (black lines) of the scattering autocorrelation $C(\tau)$ as a function of lag time $\tau$. The absolute value of the spinning frequency $\Omega$ is **d** $|\Omega| = 2.7$ Hz for left-handed circular polarization, **e** $|\Omega| = 0$ Hz for linear polarization, and **f** $|\Omega| = 2.7$ Hz for right-handed circular polarization. The absolute value of the decay constant is d $\tau_0 = 0.37$ s for left-handed circular polarization, **e** $\tau_0 = 0.11$ s for linear polarization, and **f** $\tau_0 = 0.41$ s for right-handed circular polarization.

linearly polarized light, $v_\theta$ is approximately zero, as expected (Fig. 6b). For circularly polarized light, however, we find $v_\theta$ to be different from zero: left-handed polarization results in a positive azimuthal velocity, corresponding to anti-clockwise rotation (Fig. 6a); and right-handed polarization, to negative $v_\theta$ corresponding to clockwise rotation (Fig. 6c).

This effect can be explained as follows: Due to spin angular momentum transfer from the laser light, the particle spins about its axis at frequency $\Omega$ (Fig. 6d–f). The particle's spinning motion under circular polarization is recorded via a photomultiplier. By placing a linear polarizer in front of the photomultiplier, the intensity of the scattered light changes with its orientation due to its non-sphericity. An active particle in a trap self-propels most of the time in outward direction, as rationalized by the finite polarization density $\mathbf{p} = -\nabla (u\rho)/D_r$ (Eq. (22)); the spinning motion then turns the particle axis in the azimuthal direction, $\dot{\mathbf{p}} = \Omega \times \mathbf{p}$. Solving the corresponding Fokker-Planck equation (see methods, Eq. (13)) with a finite spinning frequency, we find the azimuthal polarization $p_\theta$ given in Eq. (22) and the velocity

$$v_\theta = -\frac{\Omega u^2}{6 D_r \sqrt{D_r^2 + \Omega^2}} \frac{F}{k_B T}. \tag{7}$$

Because of the inward optical tweezers force, $F < 0$, the orbital trajectory has the same handedness as the polarized light. The azimuthal velocity is expected to vary with the third power of the beam intensity, $v_\theta \propto P(P - P_c)^2$, to vanish in the center, and to reach its maximum value at $r \approx \sigma$. Qualitatively, this expression reproduces the data of Fig. 6 with parameters corresponding to those used in Figs. 2–5. Although spin-to-orbital light momentum conversion can in principle induce similar results, we expect this effect to be comparably small. The spinning frequency $\Omega$ was obtained from fitting the scattering autocorrelation function in Fig. 6d–f with $C(\tau) = I_0^2 + 0.5 I_1^2 \exp(-\tau/\tau_0) \cos(4\pi\Omega\tau)$, where $I_0$ is the average intensity, $I_1$ the intensity fluctuation amplitude, and $\tau_0$ the decay time[34]. Surprisingly, we find that the particle is spinning under circular polarization at a frequency of about 3 Hz with a decay time of about $\tau_0 = 0.4$ s and therefore differs by 3 orders of magnitude compared to standard experiments in water[38]. Similarly, as for its reduced diffusion constant mentioned above, we expect that hydrodynamic and boundary interactions are possible causes for its much-reduced spinning motion (more details in the discussion). Regarding the much lower values of the laser power $P$ and its critical value $P_c$, note that the nanoparticles with $a = 100$ nm absorb light about ten times more than those with $a = 75$ nm, thus leading to comparable effects at a ten times weaker power. The optical tweezers potential parameter $K$ is proportional to both absorbed power and particle volume.

## Discussion

The probability density $\rho(r)$ is obtained from the stationary Fokker-Planck equation (see methods, Eq. (13)). It turns out instructive to rewrite the intermediate expression (see methods, Eq. (24)) as

$$\nabla \ln \rho = -\frac{\nabla \left( V + \frac{1}{2} H \right)}{k_B T + H}, \tag{8}$$

with $H = k_B T u^2 / \mathcal{D}^2$. For passive particles one has $H = 0$, and readily recovers the Boltzmann distribution $e^{-V/k_B T}$. The denominator of Eq. (8) may be viewed as an effective temperature. It also appears in the effective diffusion coefficient of active particles, $D_{eff} = (k_B T + H)/\gamma$[39], and the quantity $\rho H$ corresponds to the swimming pressure of active particles[40]. Assuming a constant self-propulsion velocity and discarding $k_B T$, one readily recovers the probability density $\rho \propto e^{-V/H}$ obtained previously for particles in an acoustic-wave trap[11]. From our moment expansion, however, we obtain an additional term $\frac{1}{2} H$ in the denominator of Eq. (8), which upon integration results in the intricate stationary state in Eq. (5). Since the velocity profile $u(r)$ roughly follows the laser intensity, $V + \frac{1}{2} H$ forms a Mexican hat potential which is less attractive than the bare optical tweezers potential and takes its minimum not at the beam axis but at a finite distance of the order $r_c$.

Using the experimental mean-square displacement at short times (Supplementary Fig. 4) and the measured average velocity (Fig. 3), we obtain a value for the diffusion coefficient $D = 1.09$ $\mu$m$^2$ s$^{-1}$. These numbers are smaller than the theoretical bulk Stokes-Einstein coefficient in water–2,6-lutidine $D_0 = 2.3$ $\mu$m$^2$s$^{-1}$ with the viscosity taken from ref. [24]. Similarly, the rotational diffusion coefficient used for the fit curves of Figs. 2 and 5 is smaller than the theoretical value $D_r = k_B T/(8\pi\eta a^3)$. Likewise, we would expect a spinning frequency $\Omega$ on the order of kHz and a decay constant $\tau_0$ on the order of ms for particles of similar size in water[38].

Two physical mechanism could be at the origin of this discrepancy: hydrodynamic coupling close to a solid boundary, and the confining effect of the critical droplet surrounding an active particle. First, hydrodynamic interactions increase the drag

coefficient of a sphere moving parallel to a wall[33], and similarly for rotational diffusion. In our experiment, the radiation pressure of the laser beam pushes the particle towards the glass boundary (Fig. 1), where the balance with electrostatic repulsion results in a stable vertical position close to the cover glass. This reduced separation distance has been experimentally measured for particles of similar size in ref. [22] and amounts to about 110 nm for a laser power of $P = 4$ mW, which leads to a decrease of the diffusion constant $D$ of the particle. Second, with velocities $u \sim 100\,\mu m\,s^{-1}$ and a molecular diffusion coefficient of $D_m \sim 10\,\mu m^2 s^{-1}$, the molecular Péclet number $ua/D_m$ is of the order of unity. This means that the local composition of the critical cloud, corresponding to the spinodal line of water–2,6-lutidine, does not follow the particle instantaneously but lags behind. This non-linear coupling may accelerate or slow down the particle[30]; for diffusiophoresis due to spinodal demixing, the velocity is always reduced. By the same token, the critical droplet does not follow instantaneously the particle's Brownian motion; the resulting composition gradient along the particle surface induces an opposite flow that drives the particle back and slows down diffusion. Third, the thermal conductivity contrast between the liquid and the silica wall, $\kappa_L/\kappa_W \approx \frac{1}{2}$, enhances the temperature gradient between particle and wall, resulting in a normal component of self-propulsion which could affect the diffusion coefficient[41–43].

For laser-heated gold nanoparticles in a near-critical mixture, there are two mechanisms for self-generated motion: At temperatures below the lower critical solution point (i.e., $T < T_c$), we consider thermophoresis, whereas in the opposite case (i.e., $T > T_c$), we expect diffusiophoresis to be dominant[27] (close to the lower critical point, a small change in temperature results in a large change of the spinodal composition; as a consequence, the composition gradient along the particle surface exceeds the underlying temperature gradient, thus giving rise to the surprisingly fast diffusiophoresis observed in various experiments[25].)

For spherical particles in a uniform laser field, the temperature $T(\mathbf{r})$ and the composition $\phi(\mathbf{r})$ are radially symmetric. However, active motion requires some symmetry breaking, which can in principle happen as a consequence of several possible mechanisms. First, spontaneous symmetry breaking due to a large molecular Péclet number[30] does not apply to the case of self-generated composition gradients, because Péclet numbers are too small and because composition fluctuations are not enhanced but reduced by the particle's motion. Second, the non-uniform intensity of the laser beam has little effect on gold nanoparticles, since their high thermal conductivity results in an almost isothermal surface; also, the observed velocity profile (Fig. 3) is not compatible with this mechanism because the gradient of $u$ vanishes at the center of the beam where in experiments we observe the highest value of $v$; moreover, the gradient of $u$ is only along the radial direction, but equally, fast motion is observed along the tangential component. Third, the non-spherical particle shape[44], on the contrary, turns out to be the mechanism driving our nanoparticles, as the SEM image of Fig. 1 shows a strong asphericity, and an estimate of the underlying parameters provides velocities that correspond to our experimental observations.

We have demonstrated that a nanoparticle in an optical potential in a near-critical mixture provides a model for a nanoscopic active matter system under confinement. Our system shows a strong dependence on the external confinement allowing us to control the transition from passive to active motion by tuning laser power as well as to change the orbital motion via light polarization. Our theoretical framework in comparison with our experimental observations, provides strong arguments for a propulsion mechanism grounded on the nanoparticle non-sphericity mechanism: The numerical estimate for $u$ is of the right order and magnitude, and $u$ accounts for the three observations: (i) rapid

motion in the center of the trap, (ii) rapid motion in both inward and outward direction, and (iii) rapid motion in azimuthal direction. The importance of systematic asymmetry provides insight for the future design of nanomotors. Follow-up studies could further investigate the spin-orbit coupling in combination with other types of irregular nanoparticles. In particular, nanorods due to their high aspect ratio are promising candidates characterized by much higher spin rates under circular polarization[34], improving efficiency and rotation speeds of future systems.

## Methods

**Experimental details.** We consider a suspension of gold nanoparticles (radius $a = 75 \pm 9$ nm, Sigma Aldrich) in a critical mixture of water and 2,6-lutidine at critical lutidine mass fraction $c_c = 0.286$ with a lower critical point at a temperature of $T_c \approx 34\,°C$[24] (see Supplementary Fig. 2). As shown by their SEM image in Fig. 1a, these nanoparticles possess clear crystalline faces determining their non-sphericity. The suspension is confined in a sample chamber between a microscopic slide and a coverslip with an approximate height of 100 $\mu m$.

A schematic of the experimental setup is shown in Supplementary Fig. 1. The nanoparticle's translational motion is captured via digital video microscopy at 719 Hz, whereas its spin rotation under spherical polarization is recorded by a photomultiplier (by placing a linear polarizer in front of the photomultiplier, the intensity of the scattered light changes with the particle's orientation due to its non-sphericity). An example image and trajectory of the particle is shown in Supplementary Fig. 3. The corresponding scattering intensity autocorrelation reveals oscillations with spinning frequency $\Omega$ depending on the polarization of the beam, as shown in Fig. 6d–f.

**Fokker–Planck equation.** In this section, we develop the theory for the non-equilibrium behavior observed for hot gold nanoparticles in an optical tweezers potential. We consider an active particle subject to the force $\mathbf{F} = -\nabla V$ deriving from the optical tweezers potential

$$V = -gV_0, \quad g = e^{-\frac{r^2}{2\sigma^2}} \tag{9}$$

with the depth $V_0$, the Gaussian beam profile $g$ and waist $\sigma$. Optical forces push the particle towards the solid boundary, strongly reducing the motion along the $z$-direction. Thus, we have discarded the vertical coordinate $z$, and treat the motion in the $xy$-plane only.

The equilibrium density of passive particles is determined from the steady-state condition, where motion induced by the optical tweezers force and gradient diffusion cancel each other,

$$-D\nabla\rho_{eq} + \gamma^{-1}\mathbf{F}\rho_{eq} = 0 \tag{10}$$

where $\gamma$ is Stokes' friction coefficient and $D = k_B T/\gamma$ the diffusion coefficient. With Eq. (9) this is readily integrated, resulting in the Boltzmann distribution

$$\rho_{eq} \propto e^{-V(r)/k_B T}. \tag{11}$$

This result is independent of the details of the friction coefficient. Note that $\rho_0$ cannot be normalized, since the potential takes a finite value as $r \to \infty$: a trapped particle will eventually escape after a finite residence time. As an important feature, $\rho_{eq}$ does not depend on the viscosity, since the friction factor $\gamma$ is a common factor of both terms in the steady-state condition and thus disappears. In particular, the distribution remains valid close to a solid boundary where diffusion is slowed down by hydrodynamic interactions.

The motion of an active particle in a trap arises from the gradient diffusion, the optical tweezers force $\mathbf{F}$, and the self-propulsion velocity $\mathbf{u} = u\mathbf{n}$. The direction of the latter is given by the orientation of the particle axis $\mathbf{n}$. The probability current reads accordingly

$$\mathbf{J} = -D\nabla\Psi + \gamma^{-1}\mathbf{F}\Psi + \mathbf{u}\Psi. \tag{12}$$

The probability distribution $\Psi(\mathbf{r}, \mathbf{n})$ depends on the particle position $\mathbf{r}$ and on the orientation of its axis $\mathbf{n}$, and satisfies the Fokker-Planck equation

$$\partial_t\Psi + \nabla \cdot \mathbf{J} + \mathcal{R} \cdot (\boldsymbol{\Omega} - D_r\mathcal{R})\Psi = 0, \tag{13}$$

where the last term accounts for rotational diffusion about the particle axis, with the rate constant $D_r$ and the operator $\mathcal{R} = \mathbf{n} \times \nabla_{\mathbf{n}}$, and for the angular velocity $\boldsymbol{\Omega} = \mathbf{T}/\gamma_R$ imposed by an applied torque $\mathbf{T}$. Following previous work on the dynamics of Janus particles[31,32], we resort to a moment expansion

$$\Psi(\mathbf{r}, \mathbf{n}) = \rho(\mathbf{r}) + \mathbf{n} \cdot \mathbf{p}(\mathbf{r}) + \mathbf{Q} : \left(\mathbf{nn} - \frac{1}{3}\right) + \dots \tag{14}$$

with the probability density $\rho = \langle\Psi\rangle_{\mathbf{n}}$, the polarization density $\mathbf{p} = \langle\mathbf{n}\Psi\rangle_{\mathbf{n}}$, and the quadrupolar order parameter $\mathbf{Q} = \langle(\mathbf{nn} - \frac{1}{3})\Psi\rangle_{\mathbf{n}}$, where the orientational average is defined as $\langle\dots\rangle_{\mathbf{n}} = (4\pi)^{-1}\int d\mathbf{n}(\dots)$.

The continuity relation for the former is given by

$$\partial_t \rho + \nabla \cdot \mathbf{J} = 0, \tag{15}$$

with the current

$$\mathbf{J} = -D\nabla\rho + \gamma^{-1}\mathbf{F}\rho + u\mathbf{p}. \tag{16}$$

In order to close these equations for $\rho$, we need to evaluate higher moments and to truncate this hierarchy at some order. The polarization density satisfies the continuity relation

$$\partial_t \mathbf{p} + \nabla \cdot \mathcal{J}_p + 2D_r\mathbf{p} - \mathbf{\Omega} \times \mathbf{p} = 0, \tag{17}$$

with the second-rank tensor polarization current

$$\mathcal{J}_p = -D\nabla\mathbf{p} + u\mathbf{Q} + \frac{u\rho}{3} + \frac{1}{\gamma}\mathbf{F}\mathbf{p}. \tag{18}$$

The quadrupolar order parameter $\mathbf{Q}$ is calculated for zero external torque. Putting $\Omega = 0$, we have

$$\partial_t \mathbf{Q} + \nabla \cdot \mathcal{J}_Q + 6D_r\mathbf{Q} = 0, \tag{19}$$

with the corresponding third-rank tensor current

$$\mathcal{J}_Q = -D\nabla\mathbf{Q} + \frac{2}{3}u\mathbf{p} + \frac{1}{\gamma}\mathbf{F}\mathbf{Q} + ..., \tag{20}$$

where we have discarded both the octupolar order parameter and the product $\mathbf{Qp}$.

Note that the advection term $u\rho$ in Eq. (18) generates the polarization density $\mathbf{p}$, and the advection $u\mathbf{p}$ in Eq. (20) generates the quadrupolar order parameter $\mathbf{Q}$. For small particles, rotational diffusion exceeds the derivatives of terms involving $\mathbf{p}$ and $\mathbf{Q}$.

Accordingly, we discard the current $\mathcal{J}_p$ except for the source term $u\rho$, and thus find

$$2D_r\mathbf{p} - \mathbf{\Omega} \times \mathbf{p} = -\frac{1}{3}\nabla(u\rho). \tag{21}$$

Noting that $\nabla(u\rho)$ has a radial component only and that $\mathbf{\Omega}$ is perpendicular on the plane of motion (parameterized by $r$, and $\theta$), we obtain the polarization density

$$\mathbf{p} = -\frac{\partial_r(u\rho)}{6D_r}\frac{D_r\mathbf{e}_r + \Omega\mathbf{e}_\theta}{\sqrt{D_r^2 + \Omega^2}}. \tag{22}$$

Thus, rotational diffusion favors polarization in radial direction, whereas an external spin frequency $\Omega$ turns the polarization vector in azimuthal direction. By the same token, we keep in $\mathcal{J}_Q$ the polarization advection $u\mathbf{p}$ only, and obtain

$$\mathbf{Q} = -\frac{\nabla(u\mathbf{p})}{9D_r}. \tag{23}$$

The main approximation of the above hierarchy may be viewed as an expansion in inverse powers of the rotational diffusion coefficient. Because of its variation with particle size, $D_r \propto a^{-3}$, this is justified for small enough particles.

**Non-equilibrium probability density.** The formal expression of the probability density $\rho$ is obtained from the steady-state condition for the radial component of the current, $J_r = 0$. Inserting $p_r$ and regrouping the different terms, one finds

$$\nabla\ln\rho = \frac{\mathbf{F}/\gamma - u\nabla u/6\sqrt{D_r^2 + \Omega^2}}{D + u^2/6\sqrt{D_r^2 + \Omega^2}}. \tag{24}$$

For an explicit evaluation, we have to determine the velocity $u$ as a function of the laser power. Active motion requires that the power at the particle position, $g(r)P$, exceeds the critical value $P_c$, corresponding to the lower critical temperature of water–2,6-lutidine. As the simplest relation, we take

$$u(r) = \begin{cases} C(gP - P_c) & \text{if } gP > P_c \\ 0 & \text{if } gP < P_c \end{cases}. \tag{25}$$

This describes the fact that active motion occurs only for powers above the critical value $P_c$. With the Gaussian beam profile $g = e^{-r^2/2\sigma^2}$, one readily finds that this condition is satisfied within a critical radius

$$r_c = \sigma\sqrt{2\ln(P/P_c)}. \tag{26}$$

Thus, the particle is active at distances $r < r_c$, its velocity $u(r)$ vanishes at the critical radius, and the particle is passive beyond $r_c$.

With this form, Eq. (24) is readily integrated, leading to the probability density in the active range $r < r_c$,

$$\rho(r) \propto \frac{1}{\sqrt{\mathcal{D}^2 + u(r)^2}}\exp\left(-\frac{V}{k_B T}\Phi(r)\right), \tag{27}$$

where we have defined $\mathcal{D} = \sqrt{6\sqrt{D_r^2 + \Omega^2}D}$ and

$$\Phi(r) = \frac{\mathcal{D}}{u_c + u}\arctan\frac{\mathcal{D}^2 - u_c u}{\mathcal{D}(u_c + u)}. \tag{28}$$

Beyond the critical radius $r_c$, the particle is passive ($u = 0$), and $\rho$ is given by the Boltzmann distribution $\rho_{eq} \propto e^{-V/k_B T}$. Note that in the main text, $\rho$ is discussed for $\Omega = 0$, that is, with $\mathcal{D} = \sqrt{6D_r D}$.

**Orbital velocity.** The probability and polarization densities $\rho(\mathbf{r})$ and $\mathbf{p}(\mathbf{r})$ depend on the radial coordinate only, as expected from the isotropic beam profile $g(r)$ and optical tweezers potential $V(r)$. Yet, an applied torque (for example due to angular momentum transfer from a polarized laser beam)[34,45,46] induces a spinning motion of the nanoparticle with angular velocity $\Omega$. Then, the polarization density $\mathbf{p}$ no longer points along the radial direction but acquires an azimuthal component, as shown by Eq. (22).

A finite polarization density $\mathbf{p}$ implies a mean velocity $u(\mathbf{r})\mathbf{p}(\mathbf{r})$ at position $\mathbf{r}$. In the steady state, the radial component of the corresponding current $up_r$ is compensated by the diffusion and the action of the optical tweezers force, resulting in $J_r = 0$. For the azimuthal component $J_\phi$, however, there is no such compensation force. As a consequence, a finite $p_\phi$ describes a steady orbital motion of the nanoparticle around the center of the laser beam,

$$J_\theta = p_\theta u = -\frac{\Omega}{\sqrt{D_r^2 + \Omega^2}}\frac{\partial_r(u\rho)}{6D_r}u. \tag{29}$$

At small power, one has $\partial_r(u\rho) = -u(F/k_B T)\rho$ and $\Omega \ll D_r$, resulting in the velocity

$$v_\theta = -\frac{\Omega u^2}{6D_r\sqrt{D_r^2 + \Omega^2}}\frac{F}{k_B T}. \tag{30}$$

For $\Omega = 2.7$ Hz and $u_0 = 40\ \mu m\ s^{-1}$, the azimuthal velocity is of the order of microns per second. This is in good agreement with the experimental observations reported in Fig. 6a–c.

## Data availability
The data that support the findings of this study are available from the public repository FigShare at https://doi.org/10.6084/m9.figshare.13807418.v1.

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

## Acknowledgements

We thank L. Shao for setting up the initial experiments, X. Cui for taking the SEM images of the nanoparticles, and A.A.R. Neves and R. Verre for fruitful discussions. F.S. and G.V. acknowledge partial supported by the ERC Starting Grant ComplexSwimmers (grant number 677511) and by Vetenskapsrådet (grant number 2016-03523). A.W. acknowledges support from ANR through contract Hotspot ANR-13-IS04-0003 and from ERC through contract Hiphore grant number 772725. H.S.-J., M.K., and G.V. acknowledge support from the Knut and Alice Wallenberg Foundation.

## Author contributions

F.S. conducted the experiments and analyzed the data. H.S.-J. conducted the experiments and provided Supplementary Fig. 1. A.W. worked out the theory. G.V. supervised the experiments. F.S., A.W., and G.V. wrote the manuscript. F.S., A.W., M.K., and G.V. were involved in discussions. All authors approved the final manuscript.

## Funding

## Competing interests

The authors declare no competing interests.
