## [Peer Review File · Nature Communications]

REVIEWER COMMENTS

Reviewer #1 (Remarks to the Author):

This manuscript presents a detailed experimental study and analysis of active nanoparticles confined in a harmonic potential. This work is likely to be of high interest to the community as it presents strong evidence for controllable and persistent swimming motion for nanoscale particles; in most cases this active non-equilibrium motion is 'erased' due to the high rotational diffusion coefficient of nanoscale particles. In addition to this novel finding, the authors also demonstrate that the orbital motion of the particle in the trap can be controlled by adjusting the polarization of the laser. Taken together, these findings represent a step forward in working with nanoscale particles, and are likely to inspire new active materials design in this space. Overall the manuscript is well-written and comprehensive, and is a nice conjugation of experimental, numerical, and analytic work. While this is a complex system due to the binary fluid, especially as evidence is presented that the locally de-mixed fluid cloud lags behind the particle motion, straightforward evidence is provided for all claims made in the work. I recommend this manuscript for publication if the following concerns are addressed.

1. There are significant experimental details missing, even from the supplemental information. It is unclear how the authors were able to image the 100 nm particles with enough resolution to allow for particle tracking, and which imaging mode they used. Were specialized imaging techniques used? If so, what were they? An example image of the particles, as well as small portion of the trajectories would be useful (the ones in Figure 1 are too dense to convey how well the algorithm found the particle center).
2. I am confused regarding the measurement of the diffusion coefficient. It is stated that the diffusion coefficient is determined using short-time MSD (which are not shown), but for typical active brownian particles, this part of the MSD represents the ballistic regime; it is unclear how the diffusion coefficient could be extracted from this data. Clarification on this point is needed, as this measurement is referenced several times.
3. The authors reference the reduced diffusion due to boundary effects several times. For a diffusive particle, this effect has been calculated (Faxen correction) and could be compared to the particle in this regime. The authors should have a good estimate of the surface height as they know both the laser pressure as well as the strength of the electrostatic repulsion.
4. It does not appear the authors have many statistics - each plot seems to represent the behavior of a single particle. However, as the authors state a large polydispersity of both size and shape of the particles in use, why were so few experiments done? I do not believe more statistics would change the qualitative findings presented, but I do imagine they would improve the fits shown in Fig 2 & 6.
5. The way the manuscript is written, the concept of the critical temperature is a little confusing. I had to read through a few times to realize that the particle was strongly heating, and this drove the local environment over T_c for the binary fluid. I would recommend this idea being introduced earlier - for example the caption of Figure 2 states all experiments were done at 30K below T_c , which added to my confusion. I now understand that this statement is technically true, but I find it confusing - could the estimated particle temperature be stated in the figure to add clarity?
6. The authors provide good evidence that the propulsion mechanism at work here is due to particle asymmetry. However, the best evidence is always a direct test - is there any possibility a

subset of the experiments (for example the profiles in Fig 2) could be done with spherical nanoparticles, perhaps metallic-coated silica spheres?

Reviewer #2 (Remarks to the Author):

The authors reported the study of gold nanoparticles confined by a focused laser beam in a mixture of water and 2,6-lutidine, which has a lower critical point around 34 degree Celsius. When the laser power is increased gradually, they found a cross-over of the particle probability density distribution from a Boltzmann distribution to an out-of-equilibrium. It is argued that the nanoparticles, with crystalline facets, are essentially non-spherical. Consequently, they exhibit self-propulsion when the surface temperature of the particle is larger than the critical point due to self-diffusiophoresis. The study uses the mixture of water and lutidine as a solvent which possesses a lower critical point. Such a unique solvent allows the demonstration of non-equilibrium behavior that is beyond hot Brownian motion. The experimental results are quite interesting and the analyses including examination of different possible mechanisms for active motion are comprehensive and convincing. Therefore, I recommend its publication in Nature Communication if the authors can address/clarify the following technical questions.

(1) The mechanism of diffusiophoresis for this system is quite involved, especially in considering the details of the "concentration gradient", which was very briefly mentioned in Figure 1 caption and page 5. However, the main text should provide more detailed discussion about how the concentration gradient is established, for which molecules, and the order of magnitude.

(2) Figure 2a: are there any data for nanoparticles in water at $P = 10.16$ mW? This would provide a full comparison between two different types of solvents.

(3) Where is Eq. (1) from? And how to determine the value of P_c ?

(4) Figure 3: the average velocities are determined by dividing the particle displacements between 1.39 ms, which is the resolution of the camera. The accuracy of them depends on two factors. First, would the velocities depend on the time intervals chosen otherwise, such as 4 ms or 10 ms? What does the statistical distribution of the velocities look like? Second, what is the spatial resolution of the particle trajectories? How is the resolution determined?

(5) SI page 3: while the analysis of the self-diffusiophoresis is reasonable and the authors tried to make it as quantitative as possible, a few parameters were chosen without much justification. For example, the value of ϕ_T at the bottom of page 3. Why is it equal to 5?

(6) The authors attributed the observed lower diffusion coefficient to hydrodynamic coupling to the adjacent substrate. Can the authors evaluate this effect more quantitatively and whether the separation between particle and substrate can be measured?

(7) Last sentence in the second last paragraph in page 6: Typo: nonsphericity

Reviewer #3 (Remarks to the Author):

The paper "Non-equilibrium Properties of an Active Nanoparticle in a Harmonic Potential" reports the optical trapping and manipulation of Au nanoparticles in phase-tunable liquid (which is triggered by the temperature). When increasing the laser power, the nanoparticles escapes from the center of optical trap a little bit and starts to rotate around the optical trap center.

Their observation is exciting if you haven't read their previous article (PHYSICAL REVIEW LETTERS

120, 068004 (2018)). The current results are somehow similar with that one. The major difference is that, in the present work, they used Au nanoparticles instead of silica microparticles (silica with iron-oxide inclusions). Additionally, the rotation mechanism in that paper was explained as "small asymmetries in the composition of the particle that induce asymmetries in the temperature and demixing profiles and, consequently, make the particle rotate around the optical axis". In the present work, the authors provided another theoretical model to explain the behavior of the nanoparticle, which is a result of particle's non-spherical (small asymmetries).

From my point of view, these results are correct but cover a narrow range of applicability (need a critical mixture of different mediums) and I see a lack of novelty and conceptual advance, so I do not recommend this article for publication in nature communications. Another concern: many reported papers already show optical rotation and spinning of single nanoparticles. The authors claimed the paper can provide an insight for the next generation of fast and efficient nanomotors in the introduction, but the rotation speed is very low (2.7 Hz Figure 6).

Manuscript NCOMMS-20-38791 – Response to the Reviewers' comments

We are grateful to the Reviewers for carefully reading our manuscript and for recognizing its originality, interest and quality. The points raised by the Reviewers helped us clarify very important aspects of our work and further improve our manuscript: we have now revised the text, and added more details and figures to the Supplementary Information. We provide below our point-by-point response to all the concerns raised by the Reviewers and a detailed list of the changes we have made to address them. The changes are highlighted in blue in the manuscript.

Reply to Reviewer 1:

This manuscript presents a detailed experimental study and analysis of active nanoparticles confined in a harmonic potential. This work is likely to be of high interest to the community as it presents strong evidence for controllable and persistent swimming motion for nanoscale particles; in most cases this active non-equilibrium motion is 'erased' due to the high rotational diffusion coefficient of nanoscale particles. In addition to this novel finding, the authors also demonstrate that the orbital motion of the particle in the trap can be controlled by adjusting the polarization of the laser. Taken together, these findings represent a step forward in working with nanoscale particles, and are likely to inspire new active materials design in this space. Overall the manuscript is well-written and comprehensive, and is a nice conjugation of experimental, numerical, and analytic work. While this is a complex system due to the binary fluid, especially as evidence is presented that the locally de-mixed fluid cloud lags behind the particle motion, straightforward evidence is provided for all claims made in the work. I recommend this manuscript for publication if the following concerns are addressed.

We thank the reviewer for carefully reading the manuscript and for his/her interest and positive opinion about it. In the following we reply to his/her comments in detail.

COMMENTS

- 1. There are significant experimental details missing, even from the supplemental information. It is unclear how the authors were able to image the 100 nm particles with enough resolution to allow for particle tracking, and which imaging mode they used. Were specialized imaging techniques used? If so, what were they? An example image of the particles, as well as small portion of the trajectories would be useful (the ones in Figure 1 are too dense to convey how well the algorithm found the particle center).*

We thank the reviewer for raising this important point. We image the nanoparticles with a commercial inverted microscope setup using a dark field (DF) condenser that allows only scattered light to pass through the collecting high-NA objective with 60x magnification and to be imaged onto the camera. Due to localized surface plasmon resonances (LSPR) on the nanoparticle's metallic surface, light is strongly scattered by the nanoparticle such that a bright intensity spot appears in the field of view that represents our particle (Ref. [9] of the revised Suppl. Information). In fact, thanks to the increased scattering cross section at resonance, the particle appears several orders of magnitude brighter than a dielectric nanoparticle of the same size. Therefore, the combination of DF and LSPR greatly increases the signal-to-noise ratio of metal nanoparticles. Furthermore, thanks to our microscope setup, the intensity spot of the particle appears larger (about 6x) compared to its original size, but its centroid still accurately indicates the position of the nanoparticle (see Suppl. Fig. 3a and Suppl. Video V1). To measure the fast changes in light intensity, we use a CMOS camera with high sensitivity and acquisition rates up to 1 kHz.

From the images of the particle, we track its position using an algorithm based on the radial symmetric properties of its image (see Ref. [10] in the revised Suppl. Information for more details). This particle tracking algorithm has been successfully used to track nanoscopic objects even smaller than our nanoparticles, such as single viruses (Liu *et al.*, Chem. Soc.

Rev. 2016) and proteins (Dominguez-Medina *et al.*, ACS Nano 2016). With this technique subpixel resolution down to a few nm can be achieved. The pixel-to-nanometer ratio in our experimental setup is 108.5 nm/px. As can be seen in the reconstructed trajectory (Suppl. Fig. 3b) the resolution of individual steps goes beyond the pixel resolution of the image (22×22px) and step sizes as small as few nanometers are being recorded between consecutive steps (see Suppl. Figs. 3c,d for x and y trajectory, respectively).

Following the reviewer’s suggestions, we now include in the revised manuscript an example image of our particle to show that the tracking algorithm correctly determines the particle’s center. To provide more clarification on the performance of the tracking, we now show in the revised Suppl. Information a shortened trajectory with high spatial resolution for x and y direction, respectively. We also provide now a Suppl. Video V1 showing the performance of the tracking algorithm according to Suppl. Fig. 3a.

Specifically, we have added the following section under Experimental Details in the Suppl. Information on particle tracking (page 5):

“We have imaged the nanoparticle in a commercial inverted microscope setup using a dark field condenser (see Suppl. Fig. 1). Thanks to localized surface plasmon resonances (LSPR) on the particle’s metallic surface, incident white light is strongly scattered leading to an increased scattering cross section [7]. The resulting image of the particle is visible as a bright white spot on the camera. We tracked the particle’s motion with sub-pixel resolution using the radial-symmetry particle-tracking algorithm [8]. Suppl. Fig. 3 provides an example image of the particle that appears about 6 times bigger than its actual size and indicates the center of the tracked particle (see also Suppl. Video V1), as well as reconstructed xy trajectories and the individual trajectories along x and y direction, respectively.”

We have also added Suppl. Fig. 3 as an additional figure to the revised Suppl. Information, reported here below.

Supplementary Figure 3. Performance of the tracking algorithm. **a** Example image of a nanosphere with $\alpha = 75$ nm and at fixed laser power $P = 4.36$ mW. Using the radial-symmetry particle-tracking algorithm [10], the particle’s center is precisely determined and **b** the xy trajectory reconstructed (trajectory length $t = 100$ ms). **c**, **d**

x - and y -trajectory for $t = 100$ ms, respectively. Inset: step sizes down to 5 nm. Image size 22×22 px. Scale bar represents 600 nm.

We have also added this information to the revised manuscript under methods:
“An example image and trajectory of the particle is shown in Supplementary Figure 3.”

2. I am confused regarding the measurement of the diffusion coefficient. It is stated that the diffusion coefficient is determined using short-time MSD (which are not shown), but for typical active brownian particles, this part of the MSD represents the ballistic regime; it is unclear how the diffusion coefficient could be extracted from this data. Clarification on this point is needed, as this measurement is referenced several times.

We thank the reviewer for pointing this out. Indeed, the MSD at very short timescales ($\ll 1$ μ s) represents the ballistic regime (see graphical representation of various regimes in Ref. [21] of the revised manuscript). However, due to our limited acquisition time we cannot resolve such fast movement. Instead, we have characterized the MSD at low laser power ($P = 4.36$ mW) between 1 and 20 ms lag time τ , which we found to be linear. From this fit we have determined the experimental diffusion constant D . At larger values of τ , we find the MSD to quickly reach a plateau, which characterizes the typical sub-diffusive behaviour of particles inside an optical potential. Note also that, although only few points have been taken to obtain the value of the slope, a much larger amount of data is available at short times than at larger times, making the results statistically reliable.

We now clarify this point in the revised manuscript and provide an additional figure (Suppl. Fig. 4) for the MSD and the fit of the diffusion constant D in the revised Suppl. Information (reported also here below).

The revised text on page 7 in the manuscript reads now:

“The quantity D has been calculated with a diffusion coefficient D fitted at low power $P = 4.36$ mW and on time scales larger than the inertial regime (where $\tau \ll 1$ μ s) of the trajectory mean-squared displacement between 1-20ms which we found to be linear (see Suppl. Fig. 4). Its value ($D = 1.09 \mu\text{m}^2\text{s}^{-1}$) is smaller than the bulk value in water–2,6-lutidine ($D_0 = 2.3 \mu\text{m}^2\text{s}^{-1}$, with viscosities taken from Ref. [25]).”

Supplementary Figure 4. Mean-square-displacement of radial position. Mean-square-displacement $\langle r^2 \rangle$ over lag time τ of an exemplary radial trajectory r of a nano sphere with $a = 75\text{nm}$ and at fixed laser power $P = 4.36\text{ mW}$. The dashed line is the linear fit $4D\tau$ at short time scales with $D = 1.09\ \mu\text{m}^2\text{s}^{-1}$.

- 3. The authors reference the reduced diffusion due to boundary effects several times. For a diffusive particle, this effect has been calculated (Faxen correction) and could be compared to the particle in this regime. The authors should have a good estimate of the surface height as they know both the laser pressure as well as the strength of the electrostatic repulsion.*

The reviewer raises a very important point here. The separation distance of nanoparticles with $a = 50\text{ nm}$ has been measured in experiments in Ref. [23] of the revised manuscript (Andr n *et al.*, J. Phys. Chem. C 2019). They find that depending on the laser pressure the separation distance varies and is about 110 nm for a laser power of $P = 4\text{ mW}$. Although the separation distance depends also on the strength of the electrostatic repulsion, no salts have been added to our experiments that could shield it. Following the suggestion of the reviewer, we have used the Fax n correction to calculate the diffusion constant D with the approximate separation distance given by Ref. [23], from which we find a reduced value of $D = 1.4\ \mu\text{m}^2/\text{s}$ compared to its bulk value ($D_0 = 2.3\ \mu\text{m}^2/\text{s}$). Nevertheless, our particles are slightly larger with $a = 75\text{nm}$ and therefore experience a larger radiation pressure such that we expect an even smaller separation distance and therefore smaller value of D . After carefully checking our calculations and refitting the experimental MSD (see new Suppl. Fig. 4), we find $D = 1.09\ \mu\text{m}^2/\text{s}$. Although hydrodynamic contributions between wall and particle are partly responsible for the reduction of D , we expect that the lower value found in experiments is also a result of the lateral confinement due to the critical droplet present in our experiments.

We now clarify the role of hydrodynamic interactions with the wall in the revised manuscript and provide more quantitative analysis of this effect in the discussion (page 9):

“Using the experimental mean-square displacement at short times (Suppl. Fig. 4) and the measured average velocity (Fig. 3), we obtain a value for the diffusion coefficient $D = 1.09\ \mu\text{m}^2\text{s}^{-1}$. These numbers are smaller than the theoretical bulk Stokes-Einstein coefficient in water–2,6-lutidine $D_0 = 2.3\ \mu\text{m}^2\text{s}^{-1}$ with the viscosity taken from Ref. [25].

[...]

This reduced separation distance has been experimentally measured for a particle of similar size in Ref. [23] and amounts to about 110nm for a laser power of $P = 4\text{ mW}$, which leads a to a decrease of the diffusion constant D of the particle.”

- 4. It does not appear the authors have many statistics - each plot seems to represent the behavior of a single particle. However, as the authors state a large polydispersity of both size and shape of the particles in use, why were so few experiments done? I do not believe more statistics would change the qualitative findings presented, but I do imagine they would improve the fits shown in Fig 2 & 6.*

In Figs. 2, 3 and 6 we show the results of a single particle and compare in Fig. 4 the different

results obtained from multiple particles indicating the polydispersity of particles in shape. In fact, we do not observe a qualitative change of the behaviour due to the polydispersity of the particle size. We have chosen this approach to focus the attention towards the change in particle behaviour depending on laser power. We agree with the reviewer that a larger number of particles would not change the observed behaviour and could improve error bars significantly. In fact, in Fig. 1.1 below we show the results of more particles indicating that u_0 can be found in a certain range but still varies between individual particles. In order to obtain a meaningful mean value for u_0 , averages of hundreds of particles would have to be taken in order to account for the slight differences in shape between individual particles. Each experiment including setup preparation and data analysis takes about one working day and, therefore, performing hundreds of such experiments would amount to months of work in total. Given the current circumstances and health regulations, we are unfortunately strictly limited in our ability to acquire additional large datasets.

Fig. 1.1: Results of five individual particles for the self-propulsion velocity u_0 depending on the applied laser power.

5. *The way the manuscript is written, the concept of the critical temperature is a little confusing. I had to read through a few times to realize that the particle was strongly heating, and this drove the local environment over T_c for the binary fluid. I would recommend this idea being introduced earlier - for example the caption of Figure 2 states all experiments were done at 30K below T_c , which added to my confusion. I now understand that this statement is technically true, but I find it confusing - could the estimated particle temperature be stated in the figure to add clarity?*

We thank the reviewer for providing this useful feedback. Following the reviewer's suggestion, we now clarify earlier the role of criticality in the revised manuscript and we have revised Fig. 2 and its caption.

Furthermore, we write now in the revised manuscript on page 3:

"Absorption of part of the laser light of the trapping beam heats the nanoparticle and results in a temperature profile in its vicinity. If the surface temperature exceeds T_c , a critical droplet with a modified water content $\phi(r)$ forms around the nanoparticle. Its excess surface temperature is proportional to the laser power. By choosing the critical temperature T_c ,

attained at the critical power P_c , as reference point, the excess temperature can be written as

$T(r) - T_c = \frac{a^2\beta}{3\kappa} (Pg(r) - P_c)$ with the beam profile $g(r) = e^{-\frac{r^2}{2\sigma^2}}$, the absorption coefficient β , the heat conductivity of the liquid κ , the laser power P , and the critical value P_c corresponding to the laser power at which T_c is attained.”

The caption of Fig. 2 has been adapted to indicate the increase in temperature due to the nanoparticle and reads now:

“All data are taken at environment temperature $T_0 = 3^\circ\text{C}$, i.e., ≈ 30 K below T_c . Due to strong light absorption by the particle, the particle’s surface temperature increases by 6 KmW^{-1} such that, at $P = 7.25$ and 10.16 mW, T exceeds T_c and the solution locally demixes.”

Furthermore, we have added a clear label for the particle’s surface temperature to the right panels of Fig. 2.

6. The authors provide good evidence that the propulsion mechanism at work here is due to particle asymmetry. However, the best evidence is always a direct test - is there any possibility a subset of the experiments (for example the profiles in Fig 2) could be done with spherical nanoparticles, perhaps metallic-coated silica spheres?

We thank the reviewer for posing this interesting suggestion. Ideally, this would be a great control experiment to test whether the absence of asymmetry produces any directed motion at all. Although slightly more spherical particles exist (see, e.g., Ref. Lee *et al.* ACS Nano, 2013), such particles are not commercially available and, therefore, it is impossible for us to realize the proposed experiment. As an alternative to employing perfectly spherical particles, we test the dependence of particle shape on the self-propulsion mechanism with more asymmetric particles such as nanorods, which are more readily available. We have therefore performed additional experiments using nanorods with an aspect ratio of 3:2. Our preliminary results show that the effect of self-propulsion is more pronounced and visible in the trajectories (Suppl. Fig. 5). The trajectory shows that the nanorod clearly moves around the center of the trapping beam and only rarely passes through its center (Suppl. Fig. 5a). The resulting probability distribution appears similar to that of a nanosphere at high laser powers (Suppl. Fig. 5b), whereas the velocity distributions decay quickly to a constant value suggesting that large demixing in the center pushes the particle to the periphery (Suppl. Fig. 5c).

Interestingly, if nanorods are subjected to circularly polarized light the resulting angular velocities v_θ are about 10x larger compared to our nanospheres due to their large aspect ratio (Suppl. Fig. 5d-f).

We suspect that the induced concentration fields and therefore the particle’s motion depends strongly on its orientation with respect to the center of the trapping beam. More theoretical modelling and analysis would be required to resolve the particle motion in more detail, but this is beyond the scope of this paper.

We have now added an additional section to the Suppl. Information on the dependence on particle shape where we discuss these preliminary results. The revised version reads now:

“We have further investigated the dependence of the particle’s behaviour on its shape and employed nanorods with aspect ratio of 1.5. The larger nanorods (length $l = 180$ nm, width $w = 120$ nm) already absorb enough light at $P = 1.1$ mW to induce strong demixing that pushes the particle out of the center of the trapping beam. Suppl. Fig. 5a illustrates how the particle moves in the periphery of the beam and only rarely passes through the center. The probability and velocity distributions (Suppl. Figs. 5b,c, respectively) emphasize this behaviour where a clear out-of-equilibrium signature can be found. We have further investigated the dependence of the angular velocity v_θ on circular polarization of the light beam and find similar results as those obtained for the nanosphere although at much higher speeds (about 10x larger compared to the nanosphere). This shows that particle shape plays a significant role in the resulting behaviour and that asymmetry can greatly improve rotation speeds of future nanomotors.”

We have also added a new figure (Suppl. Fig. 5) to the Suppl. Information as shown below.

Supplementary Figure 5. Dependence of particle behaviour on shape asymmetry. A nanorod with aspect ratio 1.5 (length $l = 180$ nm, width $w = 120$ nm) in a critical binary mixture and at laser power $P = 1.1$ mW shows clear out-of-equilibrium behaviour, where the trajectory in **a** shows the particle moving around the center of the trapping beam, which is reflected in **b** where the probability distribution is shifted far away from the center. The velocity distributions in **c** indicate the fast transition of the particle to the beam’s periphery where it moves at constant velocities. **d-f** show fast angular velocities under circular polarization, **d** $v_\theta > 0$ for left-handed circular polarization, and **f** $v_\theta < 0$ for right-handed circular polarization, compared to **e** where $v_\theta \approx 0$ for linear polarization. Scale bar represents 1 μm .

Reply to Reviewer 2:

The authors reported the study of gold nanoparticles confined by a focused laser beam in a mixture of water and 2,6-lutidine, which has a lower critical point around 34 degree Celsius. When the laser power is increased gradually, they found a cross-over of the particle probability density distribution from a Boltzmann distribution to an out-of- equilibrium. It is argued that the nanoparticles, with crystalline facets, are essentially non-spherical. Consequently, they exhibit self-propulsion when the surface temperature of the particle is larger than the critical point due to self-diffusiophoresis. The study uses the mixture of water and lutidine as a solvent which possesses a lower critical point. Such a unique solvent allows the demonstration of non-equilibrium behavior that is beyond hot Brownian motion. The experimental results are quite interesting and the analyses including examination of different possible mechanisms for active motion are comprehensive and convincing. Therefore, I recommend its publication in Nature Communication if the authors can address/clarify the following technical questions.

We thank the reviewer for carefully reading our manuscript and his/her interest in our work, and for recommending publication in Nature Communications. We have addressed all his/her technical questions below.

COMMENTS

- 1. The mechanism of diffusiophoresis for this system is quite involved, especially in considering the details of the “concentration gradient”, which was very briefly mentioned in Figure 1 caption and page 5. However, the main text should provide more detailed discussion about how the concentration gradient is established, for which molecules, and the order of magnitude.*

We thank the reviewer for pointing this out and providing valuable feedback. The subtle point is that, for constant surface composition, the local composition gradient is proportional to the local curvature. For a non-spherical particle, this implies a component of the composition gradient parallel to the surface. In other words, the usual mechanism of self-diffusiophoresis does not apply to homogeneous colloidal spheres, since their symmetry does not allow a composition gradient along the surface. Therefore, we propose self-propulsion that arises from the non-spherical shape of our nanoparticles, visible in Fig. 1a. The large thermal conductivity of gold imposes an isothermal surface at the particle surface, yet the temperature gradient away from the particle surface varies with the local curvature. Thus, above the critical point, the composition $\phi(r)$ at constant distance varies along the particle surface, and the parallel component of the gradient $\nabla_{\parallel}\phi$ induces creep flow and self-propulsion of the particle.

Following the reviewer’s suggestion, we now explain the concentration gradient in more detail in the revised manuscript on page 4:

“With increasing power, the particle’s surface temperature exceeds the lower critical point T_c of water-2,6-lutidine (see SI [26]), causing a local modification of the composition according to the spinodal line of the phase diagram. Then, the particle is surrounded by a droplet of modified water content, $\phi(r) - \phi_c \propto \sqrt{T(r) - T_c}$, where the sign of the excess term depends

on the wetting properties of the surface. Within this droplet, isotherms correspond to iso-composition surfaces. For non-uniformly heated particles, the resulting composition gradient parallel to the surface, $\nabla_{\parallel}\phi$, drives self-diffusiophoresis. Indeed, active motion above T_c has been reported for both laser-heated Janus particles [27, 28] and silica colloids with iron-oxide inclusions [16]. This mechanism was worked out in detail by analytical theory [29] and simulations [30].”

and on page 5:

“Starting from an axisymmetric profile $R(\theta) = a(1 + \chi(\theta))$ with $\chi = n\alpha_n P_n(\cos\theta)$, with the polar angle θ and Legendre polynomials P_n , and evaluating the temperature profile in the vicinity of the isothermal surface of a gold particle, we obtain self-diffusiophoresis at a velocity $u \propto \alpha^2 = \sum_{n=2}^{\infty} \frac{3n+2}{2n+3} \alpha_n \alpha_{n+1}$. Thus, motion arises from the $2n + 3$ superposition of odd and even Fourier components of the particle shape. The series starts at $n = 2$, since the dipolar term $n = 2$ corresponds to an irrelevant displacement. For our fits, we assume that less than eight modes contribute with $\alpha_n \sim 0.1$ and thus find $\alpha^2 \sim$ a few percent.”

In the revised Suppl. Information (pages 3-4) we also discuss the self-propulsion velocity in more detail:

“Previous experiments show that self-propulsion of Janus particles in water-lutidine above T_c is much faster than in water [1,2], implying that the parameter ϕ_T is much larger than unity. This is confirmed by a mean-field treatment of the spinodals above T_c [3] and numerical simulations [4]. That is why we neglect the thermophoretic contribution (S.9). With the notable exception of electric double-layer forces [5], the thermal and chemical surface forces driving active particles cannot be calculated quantitatively; the above numbers for the parameters $H_w - H_l, \sim k_B T, \alpha^2 \sim 0.06$ and $\phi_T \sim 5$ are reasonable estimates, in line with previous experiments.”

2. Figure 2a: are there any data for nanoparticles in water at $P = 10.16$ mW? This would provide a full comparison between two different types of solvents.

We currently do not possess data for water at $P = 10.16$ mW but we don't expect any change in behaviour apart from an increase in the trapping stiffness of the laser beam and thus an even narrower probability distribution, similarly to what can be observed in the transition from $P = 4.36$ mW to 7.25 mW. In the case of water, the particle is only subjected to optical forces such that its probability distribution follows a Boltzmann curve. This confining effect of the optical potential on nanoparticles for different laser powers has already been observed and explained in more detail in Ref. [23] of the revised manuscript. We agree, however, that in principle an additional panel with $P = 10.16$ mW could allow for better visualization and comparison with the water-2,6-lutidine data. Given the current circumstances imposed by health authorities, access to labs is strictly limited and thus we are not able to acquire new data in a reasonable time frame.

We now clarify this point in the revised manuscript on page 3:

“Since the stiffness of the potential increases with laser power, the distribution function is narrower at larger P . Consequently, an even narrower distribution function is expected at larger laser powers (e.g., $P = 10.16$ mW).”

3. Where is Eq. (1) from? And how to determine the value of P_c ?

Eq. (1) expresses that the excess temperature is proportional to the applied power, $T - T_0 \propto P$. In our case it is more convenient to rewrite this relation with respect to the values at the critical point, that is $T - T_c \propto P - P_c$. We have rewritten this passage in the revised manuscript (page 3):

“Absorption of part of the laser light of the trapping beam heats the nanoparticle and results in a temperature profile in its vicinity. If the surface temperature exceeds T_c , a critical droplet with a modified water content $\phi(r)$ forms around the nanoparticle. Its excess surface temperature is proportional to the laser power. By choosing the critical temperature T_c , attained at the critical power P_c , as reference point, the excess temperature can be written as

$T(r) - T_c = \frac{a^2\beta}{3\kappa}(Pg(r) - P_c)$ with the beam profile $g(r) = e^{-\frac{r^2}{2\sigma^2}}$, the absorption coefficient β , the heat conductivity of the liquid κ , the laser power P , and the critical value P_c corresponding to the laser power at which T_c is attained”.

We also indicate explicitly that P_c is the power required to heat the particle from the sample temperature T_0 to the critical temperature T_c of water-lutidine. We determine the value of P_c by fitting Eqn. 5 in the revised manuscript to the propulsion velocity data u_0 plotted in Fig. 5. As there is no self-propulsion below P_c , its value is determined by the cross-section of the fit with the x -axis.

We now also emphasize this point in the main text of the revised manuscript on page 6: “The three particles have the same radius a and absorption coefficient β ; accordingly, they experience the same optical tweezers potential and reach the critical point at the same laser power P_c (obtained from the fit of u_0 using Eq. (5))”

4. Figure 3: the average velocities are determined by dividing the particle displacements between 1.39 ms, which is the resolution of the camera. The accuracy of them depends on two factors. First, would the velocities depend on the time intervals chosen otherwise, such as 4 ms or 10 ms? What does the statistical distribution of the velocities look like?

We thank the reviewer for posing this useful question. As the reviewer correctly points out, the resolution of the camera is 1.39 ms and thus our time resolution is determined by the camera acquisition speed. This time resolution is sufficient because both the the rotational diffusion time and the trap characteristic time are larger than our time resolution.

Considering longer time steps (e.g., $dt = 4.2, 9.8\text{ms}$) would introduce a bias in the estimation of the velocities, as can be seen in Fig. 2.1.

Fig.2.1: Comparison of the time interval of acquisition (1.4, 4.2, 9.7 ms) depending on the velocity distribution for the total velocity v , the absolute radial velocity v_r , and the absolute angular velocity v_θ . Data for $P = 10.16$ mW are taken.

Second, what is the spatial resolution of the particle trajectories? How is the resolution determined?

The spatial resolution of the particle trajectories is determined by our optical setup. In order to visualize our particle we exploit its strong scattering properties under bright field illumination. We then image the particle using an objective with 60x magnification, which is then projected onto our camera. Using a radial particle symmetry tracking we obtain sub-pixel resolution on the order of several nanometres, which we now also show in a new Suppl. Fig. 3. More details on the experimental setup and the tracking algorithm can be found in the response to reviewer 1 comment 1 and in the revised Suppl. Information under the section “Particle Tracking”.

We have added this information now in the revised Suppl. Information on page 5:

“We have imaged the nanoparticle in a commercial inverted microscope setup using a dark field condenser (see Suppl. Fig. 1). Due to localized surface plasmon resonances (LSPR) on the particle’s metallic surface, incident white light is strongly scattered leading to an increased scattering cross section [9]. The resulting image of the particle is visible as bright white spot on the camera. We tracked the particle’s motion with sub-pixel resolution using radial symmetry calculations [10]. Suppl. Fig. 3 provides an example image of the particle that appears about 6 bigger than its actual size and indicates the center of the tracked particle (see also Suppl. Video V1), as well as reconstructed xy trajectories and the individual trajectories along x and y direction, respectively.”

5. SI page 3: while the analysis of the self-diffusiophoresis is reasonable and the authors tried to make it as quantitative as possible, a few parameters were chosen without much justification. For example, the value of ϕ_T at the bottom of page 3. Why is it equal to 5?

Quite generally, the absolute value of the velocity of active particles cannot be derived quantitatively, mainly because of the intricate physics of surface forces acting parallel to the surface. We have rewritten the discussion on page 3 of the SI and point out two important aspects: First, previous experiments on active particles in water-lutidine indicate clearly that the diffusiophoresis dominates with respect to thermophoresis, in other words ϕ_T is larger than unity. Second, the numbers used for our fits involve only one overall factor, consisting of

the van der Waals forces $H_w - H_l$, the anisotropy parameter α^2 , and the factor ϕ_T . In our discussion we now show that the values taken for these parameters are reasonable and in line with previous experiments [Buttinoni *et al.* J. Phys. Condens. Matter 2012 and Lozano *et al.* New J. Phys. 2018].

We have corrected this now in the revised Suppl. Information on pages 3-4:

“Previous experiments show that self-propulsion of Janus particles in water-lutidine above T_c is much faster than in water [1, 2], implying that the parameter ϕ_T is much larger than unity. This is confirmed by a mean-field treatment of the spinodals above T_c [3] and numerical simulations [4]. That is why we neglect the thermophoretic contribution (S.9). With the notable exception of electric double-layer forces [5], the thermal and chemical surface forces driving active particles cannot be calculated quantitatively; the above numbers for the parameters $H_w - H_l, \sim k_B T, \alpha^2 \sim 0.06$ and $\phi_T \sim 5$ are reasonable estimates, in line with previous experiments [1, 2].”

6. The authors attributed the observed lower diffusion coefficient to hydrodynamic coupling to the adjacent substrate. Can the authors evaluate this effect more quantitatively and whether the separation between particle and substrate can be measured?

The distance between particle and surface has already been experimentally measured by our co-authors using total internal reflection microscopy (TIRM) for particles of similar size and shape in water and in a similar experimental setup (Ref. [23] of the revised manuscript). The distance depends on two parameters: the radiation pressure and the ionic concentration shielding the electrostatic repulsion. Assuming that there no spurious ions are present, the measured separation distance for a laser power of $P = 4$ mW is about 110 nm. Using the Faxen formula and the viscosity of water-2,6-lutidin (taken from Ref. [25] of the revised manuscript), a reduced diffusion constant $D = 1.4 \mu\text{m}^2/\text{s}$ is found, compared to bulk measurements with $D_0 = 2.3 \mu\text{m}^2/\text{s}$. We have carefully revised the calculations of our experimental D using the fitting of the MSD at low power and find a corrected value of $D = 1.09 \mu\text{m}^2/\text{s}$ (see Suppl. Fig. 4). This shows that indeed hydrodynamic interaction partly explains the reduction in D but further effects such as the confining effect of the droplet have to be taken into account as well.

We now added this information to the discussion on page 9 of the revised manuscript:

“In our experiment, the radiation pressure of the laser beam pushes the particle towards the glass boundary (Fig. 1), where the balance with electrostatic repulsion results in a stable vertical position close to the cover glass. This reduced separation distance has been experimentally measured for a particle of similar size in Ref. [23] and amounts to about 110 nm for laser powers of $P = 4.36\text{mW}$, which leads a to a decrease of the diffusion constant D .”

We now also discuss the effect of thermal conductivity contrast on the confinement in the revised manuscript on page 9:

“Third, the thermal conductivity contrast between the liquid and the silica wall, $\frac{\kappa_L}{\kappa_W} \approx \frac{1}{2}$,

enhances the temperature gradient between particle and wall, resulting in a normal

component of self-propulsion which could affect the diffusion coefficient [43-45].”

7. Last sentence in the second last paragraph in page 6: Typo: nonsphericity

We thank the reviewer for pointing out this typo. We have corrected it.

Reply to Reviewer 3:

The paper “Non-equilibrium Properties of an Active Nanoparticle in a Harmonic Potential” reports the optical trapping and manipulation of Au nanoparticles in phase-tunable liquid (which is triggered by the temperature). When increasing the laser power, the nanoparticles escape from the center of optical trap a little bit and start to rotate around the optical trap center.

Their observation is exciting if you haven't read their previous article (PHYSICAL REVIEW LETTERS 120, 068004 (2018)). The current results are somehow similar with that one. The major difference is that, in the present work, they used Au nanoparticles instead of silica microparticles (silica with iron-oxide inclusions). Additionally, the rotation mechanism in that paper was explained as “small asymmetries in the composition of the particle that induce asymmetries in the temperature and demixing profiles and, consequently, make the particle rotate around the optical axis”. In the present work, the authors provided another theoretical model to explain the behavior of the nanoparticle, which is a result of particle's non-spherical (small asymmetries).

From my point of view, these results are correct but cover a narrow range of applicability (need a critical mixture of different mediums) and I see a lack of novelty and conceptual advance, so I do not recommend this article for publication in nature communications. Another concern: many reported papers already show optical rotation and spinning of single nanoparticles. The authors claimed the paper can provide an insight for the next generation of fast and efficient nanomotors in the introduction, but the rotation speed is very low (2.7 Hz Figure 6).

We thank the reviewer for sharing his/her criticisms and point of view. Below we reply to all points raised, providing a clearer explanation of the novelty of this work when compared to previously published results.

We agree with the reviewer that there are some similarities with our previous work as we employ active particles propelled by self-diffusiophoresis inside an optical potential. However, our nanoparticle presented here behaves strikingly different compared to microparticles, as random fluctuations on the nanoscale are non-negligible. This implies that the physics behind the nanoparticle system and of the microparticle system are substantially different. In fact, directed motion is strongly impeded as rotational Brownian motion occurs on much shorter times that scale with its volume (a^{-3}). This implies that a 10x smaller nanoparticle rotates on time scales 1000x faster than its microscopic equivalent, which in contrast moves continuously on fixed circular orbits and at constant frequency. Consequently, the rotational motion of a nanoparticle cannot be observed directly differently from rotating microparticles (Refs. [1,2] of the revised Suppl. Information and Ref. [16] of the revised manuscript). For these reasons, we believe that this manuscript provides some novel insight into the behaviour of active nanoscopic particles subjected to harmonic potentials.

When starting our experiments with metallic nanoparticles we have been surprised by our initial results that clearly demonstrated an out-of-equilibrium signature and therefore

indicated self-propulsion of the particle. These results are non-trivial, because metal nanoparticles in contrast to dielectric microparticles possess an isothermal surface such that no temperature gradient across their surface can be responsible for their directed motion. Only after investigating the accidental non-sphericity did we manage to identify and model this important effect.

Although other spinning nanomotors have been proposed (e.g., Refs. [4, 18-21] of the revised manuscript), their motion does not reflect self-propulsion. Instead they are subjected to a rotating electro-magnetic fields and whose activity reflects a hot Brownian motion only. The reviewer is right that the reported rotation frequency is low, as our nanoparticles only slightly deviate from a perfect sphere, which we think make these results even more impressive given that small changes in shape can result into total velocities above 100 $\mu\text{m/s}$. To emphasize this point even more, we now also include preliminary data of highly nonspherical nanoparticles, i.e. nanorods, that show much higher values of its angular velocity ($v_\theta \approx 100\mu\text{m}$) compared to a nanosphere and whose spinning can be controlled with circularly polarized light.

We have now added another section in the Suppl. Information on the dependence on particle shape to emphasize this point:

“We have further investigated the dependence of the particle’s behaviour on its shape and employed nanorods with aspect ratio of 1.5. The larger nanorods (length $l = 180\text{nm}$, width $w = 120\text{nm}$) already absorb enough light at $P = 1.1 \text{ mW}$ to induce strong demixing that pushes the particle out of the center of the trapping beam. In Suppl. Fig. 5a illustrates how the particle moves in the periphery of the beam and only rarely passes through the center. The probability and velocity distributions (Suppl. Figs. 5b,c, respectively) emphasize this behaviour where a clear out-of-equilibrium signature can be found. We have further investigated the dependence of the angular velocity v_θ on circular polarization of the light beam and find similar results as for our nanosphere although at much higher speeds (about 10x larger compared to the nanosphere). This shows that particle shape plays a significant role in the resulting behaviour and where asymmetry can greatly improve rotation speeds of future nanomotors.”

REVIEWERS' COMMENTS

Reviewer #1 (Remarks to the Author):

I have carefully looked over the revised draft, as well as the authors' point-by-point response. As I said in my first review, I believe this manuscript will be of high interest to the community, and it represents a new step forward that is likely to inspire advances in the design of nanoscale active matter systems. The authors' have addressed all of my concerns with the manuscript, and I recommend it for publication with no reservations.

Reviewer #2 (Remarks to the Author):

The authors have addressed the reviewers' comments satisfactorily. Therefore, I recommend its publication in Nature Communication.

Reviewer #3 (Remarks to the Author):

I believe there are some interesting findings from this paper that could be useful to the optical nano-manipulation community. I appreciate the additional section to introduce more details to this paper. I am still of the opinion that the original contributions of this paper are unsatisfactory but if the editors and other reviewers disagree then I don't feel the need to make a strong stand on this point.

Manuscript NCOMMS-20-38791a – Response to the Reviewers' comments

We are grateful to the Reviewers for providing valuable feedback and comments to our original manuscript and thank them for recommending the revised manuscript for publication in Nature Communications.

Reviewer #1 (Remarks to the Author):

I have carefully looked over the revised draft, as well as the authors' point-by-point response. As I said in my first review, I believe this manuscript will be of high interest to the community, and it represents a new step forward that is likely to inspire advances in the design of nanoscale active matter systems. The authors' have addressed all of my concerns with the manuscript, and I recommend it for publication with no reservations.

We thank the reviewer for his/her original comments and carefully reading our revised version of the manuscript.

Reviewer #2 (Remarks to the Author):

The authors have addressed the reviewers' comments satisfactorily. Therefore, I recommend its publication in Nature Communication.

We are grateful for his/her valuable feedback that helped us to improve our manuscript and thank him/her for recommending it for publication in Nature Communication.

Reviewer #3 (Remarks to the Author):

I believe there are some interesting findings from this paper that could be useful to the optical nano-manipulation community. I appreciate the additional section to introduce more details to this paper. I am still of the opinion that the original contributions of this paper are unsatisfactory but if the editors and other reviewers disagree then I don't feel the need to make a strong stand on this point.

We thank the reviewer for his/her interest in the paper and in the additional information we have made to provide a better picture of the novelty of our work.